# Autocatalytic photoredox Chan-Lam coupling of free diaryl sulfoximines with arylboronic acids

Cong Wang[1,6], Hui Zhang[1,6], Lucille A. Wells[2,6], Tian Liu[3], Tingting Meng[1], Qingchao Liu[3], Patrick J. Walsh [2], Marisa C. Kozlowski [2✉] & Tiezheng Jia [1,4,5✉]

*N*-Arylation of *NH*-sulfoximines represents an appealing approach to access *N*-aryl sulfoximines, but has not been successfully applied to *NH*-diaryl sulfoximines. Herein, a copper-catalyzed photoredox dehydrogenative Chan-Lam coupling of free diaryl sulfoximines and arylboronic acids is described. This neutral and ligand-free coupling is initiated by ambient light-induced copper-catalyzed single-electron reduction of *NH*-sulfoximines. This electron transfer route circumvents the sacrificial oxidant employed in traditional Chan-Lam coupling reactions, increasing the environmental friendliness of this process. Instead, dihydrogen gas forms as a byproduct of this reaction. Mechanistic investigations also reveal a unique autocatalysis process. The C–N coupling products, *N*-arylated sulfoximines, serve as ligands along with *NH*-sulfoximine to bind to the copper species, generating the photocatalyst. DFT calculations reveal that both the *NH*-sulfoximine substrate and the *N*-aryl product can ligate the copper accounting for the observed autocatalysis. Two energetically viable stepwise pathways were located wherein the copper facilitates hydrogen atom abstraction from the *NH*-sulfoximine and the ethanol solvent to produce dihydrogen. The protocol described herein represents an appealing alternative strategy to the classic oxidative Chan-Lam reaction, allowing greater substrate generality as well as the elimination of byproduct formation from oxidants.

[1] Shenzhen Grubbs Institute and Department of Chemistry, Southern University of Science and Technology, Shenzhen, Guangdong, China. [2] Roy and Diana Vagelos Laboratories, Penn/Merck Laboratory for High-Throughput Experimentation, Department of Chemistry, University of Pennsylvania, Philadelphia, PA, USA. [3] Department of Pharmaceutical Engineering, College of Chemical Engineering, Northwest University, Xi'an, Shanxi, China. [4] State Key Laboratory of Elemento-Organic Chemistry, Nankai University, Tianjin, China. [5] Guangdong Provincial Key Laboratory of Catalysis, Shenzhen, Guangdong, China. [6]These authors contributed equally: Cong Wang, Hui Zhang, Lucille A. Wells. ✉email: jiatz@sustech.edu.cn; marisa@sas.upenn.edu

Not long ago, sulfoximines were virtually unknown structural motifs in biological and medicinal chemistry[1–5]. This changed with Bayer's finding of the sulfoximine-based pan-CDK inhibitor (BAY 1000394, Fig. 1a)[6]. Since then, sulfoximines have received increasing attention as promising scaffolds and building-blocks in organic synthesis[3,4] and as synthetic bioactive molecules in drug development (Fig. 1a)[7,8]. Sulfoximines also find widespread applications in agricultural chemistry, as exemplified by sulfoxaflor, a marketed insecticide[9]. Other uses of sulfoximines are as ligands in asymmetric catalysis[4,10,11].

Considering the unique structural features of sulfoximines, and their successful applications in medicinal chemistry, much effort has been devoted to their preparation. Conventional approaches to sulfoximines involve oxidation of sulfilimines, oxidative imination of sulfoxides[5], or direct sequential oxidation of sulfides to sulfoximines[12,13]. These methods require strong oxidizing agents and are less suitable for late stage modifications. Direct arylation of the nitrogen of NH-sulfoximines represents an appealing and efficient approach to N-arylated sulfoximines[14–17]. The Buchwald–Hartwig amination was first applied to arylation of alkyl aryl NH-sulfoximines by Bolm and co-workers (Fig. 1b)[18]. Surprisingly, general methods for the N-arylation of diaryl NH-sulfoximines, Ar₂SO(NH), using the Buchwald–Hartwig approach have not been developed to date. This gap is attributed, in part, to the lower nucleophilicity of these species relative to alkyl substituted NH-sulfoximines, (alkyl)ArSO(NH). The only successful examples of arylations of diaryl NH-sulfoximines [Ar₂SO(NH)] employed diaryliodonium salts (two examples)[19] or aryl tosylates (three examples)[20] as coupling partners. Although these advances are of significance, the scope they define is quite narrow and currently unsuitable for generating diversity.

The first systematic study of the arylation of diaryl NH-sulfoximines was disclosed by the Yu group (Fig. 1b)[21]. Their optimal conditions employed 40 mol % Cu(acac)₂ and benzoyl peroxide at 130 °C to drive N-arylation via a radical pathway. König and Wimmer and the team of Hog, Meier, and co-workers, described similar photoredox-catalyzed N-arylation of free

sulfoximines with electron-rich arenes (Fig. 1b)[22–24]. Although elegantly circumventing the need for an electrophilic coupling partner, the application of this approach is hampered by limited substrate scope and regioselectivity issues arising from the reaction of highly reactive radical intermediates. Several scattered examples of NH-diaryl sulfoximine N-arylations were described by Bolm[25] and other groups[20,24,26,27]. Unfortunately, low yields were generally observed. A direct and efficient N-arylation protocol for NH-diaryl sulfoximines with broad substrate scope, including heteroaryl coupling partners, is still an unmet need.

The copper-catalyzed Chan-Lam amination[28,29] is one of the most powerful tools to construct C–N bonds[30], offering the advantages of (a) inexpensive copper-based catalysts that can often operate without added ligands, (b) mild reaction conditions, and (c) use of readily available boronic acids as coupling partners. Because the Chan-Lam amination is a coupling of two nucleophiles, an oxidant is required. Thus, a major drawback of Chan-Lam aminations arise from the oxidation of the substrates by the oxidant, including oxidative deboration of arylboronic acids to phenols[31,32]. In this regard, the discovery of a Chan-Lam coupling that does not require addition of an external oxidant would represent a significant advance.

The role of the oxidant in Chan-Lam couplings is generally believed to be to oxidize Cu(I) to Cu(II)[31,33], although it has also been postulated that it oxidizes Cu(II) to Cu(III)[26]. Inspired by the pioneering work of Fu, Peters, and their co-workers on the photo-induced Ullmann coupling[34,35] and related C(sp³)–N bond constructions[36–39], we hypothesized that the photo-induced single-electron oxidation of the catalysts from Cu(I) to Cu(II) could serve as an alternative to external oxidants, allowing the development of a photoredox Chan-Lam coupling protocol without external oxidants. Of note, Kobayashi and co-workers[40] reported a visible-light-mediated Chan-Lam coupling of arylboronic acids with aniline derivatives. This study, which employed an Ir-photocatalyst with dioxygen as the terminal oxidant, demonstrated the compatibility of Chan-Lam amination with photoredox conditions.

Another source of inspiration came from the Bolm group, who developed a copper-catalyzed Chan-Lam amination of

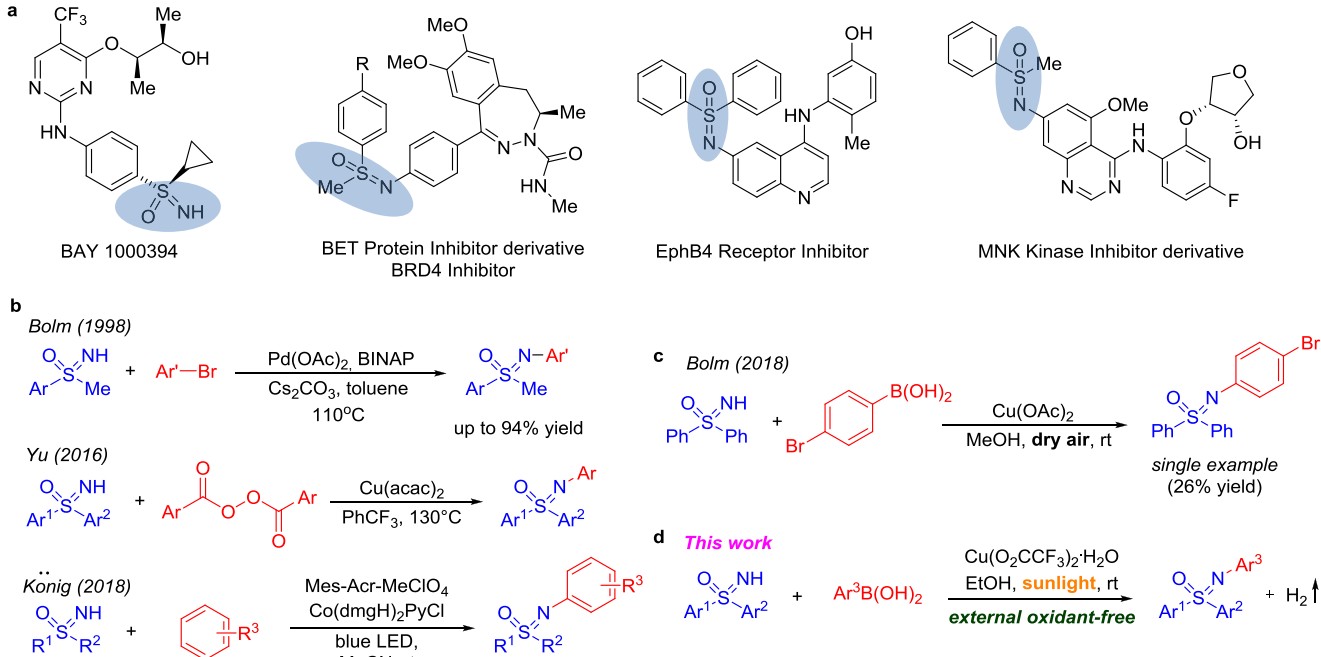

**Fig. 1 N-Arylation of sulfoximines. a** Representative examples of bioactive sulfoximines. **b** Three reported strategies to N-aryl sulfoximines. **c** N-arylation of diaryl sulfoximine via traditional Chan-Lam reaction. **d** This photoredox Chan-Lam Coupling of free diaryl sulfoximines with arylboronic acids.

*NH*-dialkyl- and *NH*-arylalkyl sulfoximines with arylboronic acids employing oxygen in dry air as the oxidant[14,25]. Their single example of an *N*-arylation employing 4-bromophenylboronic acid and *NH*-diphenyl sulfoximine (26% yield, Fig. 1c) attracted our attention. Herein, we report the development of a general and straightforward light-induced copper-catalyzed Chan-Lam arylation of free diaryl sulfoximines with arylboronic acids (Fig. 1d). Notable features of this protocol include: (a) a variety of functional groups are well tolerated, including heteroarylboronic acids and heteroaryl-containing sulfoximines; (b) no external oxidant is required, eliminating oxidant-based byproduct formation; (c) an inexpensive copper source serves as photocatalyst; (d) mechanistic studies reveal an autocatalysis process, arising from ligation of *N*-aryl sulfoximines to copper(I) species; and (e) dihydrogen is the byproduct in this coupling, which is simply purged from the reaction vessel.

## Results and discussion

**Reaction discovery and optimization**. Our goal was to develop a Chan-Lam coupling that did not require external oxidant. We initiated the optimization of copper-catalyzed *N*-arylation of *NH*-diphenyl sulfoximine **1a** (1.0 equiv) with phenylboronic acid **2a** (2.3 equiv) in methanol (0.3 M) for 48 h with catalytic Cu(O$_2$CCF$_3$)$_2$·H$_2$O (20 mol %), similar to the conditions reported by Bolm and co-workers[25] (Fig. 1c). Rather than conducting the reaction under oxidizing conditions, however, it was performed under an argon atmosphere to exclude the sacrificial oxidant (O$_2$ in air). We observed formation of the desired product in 25% AY of **3aa** (AY = assay yield, Table 1, entry 1). After a survey of copper sources, Cu(O$_2$CCF$_3$)$_2$·H$_2$O was determined to be the best among those examined, giving 36% AY of **3aa** (entry 4 vs. entries 1–3, Table 1). We next examined Cu(O$_2$CCF$_3$)$_2$·H$_2$O in commonly used alcohol solvents (EtOH, *i*PrOH, and *t*BuOH, entries 5–7, Table 1). When EtOH was employed, **3aa** was afforded in 90% AY (entry 5, Table 1). We next examined the catalyst loading and found that decreasing to 10 mol % Cu(O$_2$CCF$_3$)$_2$·H$_2$O, provided 90% AY of **3aa** (entry 8, Table 1). Other reaction parameters, such as reagent ratio and concentration, were next examined. The stoichiometry of boronic acid **2a** could be decreased from 2.3 to 2.0 equiv, without impacting the AY (90%, entry 10, Table 1). Interestingly, the concentration could be increased from 0.3 to 1.5 M, generating **3aa** in 95% AY (entry 12, Table 1) and 94% isolated yield. No coupling product formed in the absence of Cu(O$_2$CCF$_3$)$_2$·H$_2$O, confirming that the copper acted as catalyst. Therefore, the optimized conditions for the *N*-arylation were determined to be: sulfoximine **1a** as limiting reagent, boronic acid **2a** (2.0 equiv), Cu(O$_2$CCF$_3$)$_2$·H$_2$O (10 mol %) as catalyst, in ethanol (1.5 M) at room temperature for 48 h. Other boronic acid equivalents were also explored. The pinacol ester of phenylboronic acid led to poor yield of **3aa**, presumably owing to inhibition of the copper by coordination to pinacol (entry 15, Table 1)[31]. Furthermore, the employment of three equiv B(OH)$_3$ as additive[31] exerted negligible effect on the outcome of the transformation, and only 20% yield of **3aa** was obtained (entry 15, Table 1). Phenyl trifluoroborate was only effective in forming **3aa** at reduced levels (entry 16, Table 1).

**Substrate scope**. With the optimized conditions in hand (Table 1, entry 12), we investigated the substrate scope of the arylboronic acids in the copper-catalyzed arylation of **1a** (Table 2). Boronic acids bearing electron-donating 4-Me and 4-OMe groups (**2b**, **2c**) produced the corresponding arylated sulfoximines **3ab** and **3ac** in 96% and 85% yield, respectively. Halogenated arylboronic acids containing 4-F (**2d**), 4-Cl (**2e**) or 4-Br (**2f**) substitution reacted smoothly, generating **3ad–f** in 83–91% yield. Sterically hindered

1-naphthylboronic acid (**2g**) afforded 75% yield **3ag** with 15 mol % copper catalyst. 3,5-Dimethylphenylboronic acid (**2h**) also proved to be a good coupling partner, and **3ah** was obtained in 80% yield. The chemistry was also accommodated by arylboronic acids possessing strongly electron-withdrawing groups to furnish **3ai–3ak** in yields ranging from 59% to 82%.

The oxidant-free conditions enabled use of a broad range of functional groups on the arylboronic acids, including aldehyde (**2l**), methyl ketone (**2m**), methyle ester (**2n**), and even free hydroxyl (**2o**) groups (60–80%). The coupling of the unprotected phenol is particularly impressive, because phenols are well known to couple in Chan-Lam reactions to give diaryl ethers[27]. Heterocyclic sulfoximines exhibit a broad range of biological activities and, thus, are important synthetic targets. To our delight, 3-pyridinyl (**2p**), 3-thiophenyl (**2q**), and 5-*NH*-indolyl (**2r**) boronic acid derivatives furnished the corresponding products **3p-r** in 65–75%. Of note, our arylation displayed excellent chemoselectivity favoring arylation of the sulfoximine **3ar** over arylation of the indole *N*–H. In a preliminary probe, vinylboronic acid failed to form the desired product under the standard conditions.

Next, the substrate generality of *NH*-diaryl sulfoximines was investigated using phenylboronic acid (**2a**) (Table 2). Phenyl and electron-donating 4-Me (**1b**) and 4-OMe (**1c**) 3,5-Me (**1d**) substituted sulfoximines reacted smoothly with **2a** to furnish **3aa**–**3da** in 75–95% yield. The sulfoximine possessing a 4-NHAc group (**1e**) furnished the product **3ea** yield 63% with no observed *N*-arylation of the amide. *NH*-Sulfoximines possessing halogens and electron-withdrawing functionality, such as 4-F (**1f**), 4-Cl (**1g**), 4-CN (**1h**), 4-CF$_3$ (**1i**), 4-NO$_2$ (**1j**) or 4-CO$_2$Me (**1k**) groups, proved to be compatible coupling partners, providing **3fa**–**3ka** in 71–92% yields. A free sulfoximine bearing 4-COMe (**1l**) provided the arylation product **3la** in 80% yield. Furthermore, unprotected 4-COOH (**1m**) and 3–OH (**1n**) appended phenyl sulfoximines proceeded smoothly under the optimized conditions to provide **3ma** and **3na** in 88% and 82%, respectively. These results highlight the excellent chemoselectivity of our protocol, favoring C-N bond coupling over C-O bond formation. Sterically hindered 1-naphthyl phenyl sulfoximine **1o** required an extended reaction time (60 h), but furnished coupling product **3oa** in 70% yield. Notably, heterocyclic 2-pyridyl (**1p**) and 3-thiophenyl sulfoximines (**1q**) were also suitable substrates, affording **3pa** and **3qa** in 65% and 73% yield, respectively.

To demonstrate the utility of our copper-catalyzed *N*-arylation protocol, the bis-*NH*-sulfoximine (**1r**) furnished the bis-arylation product **3ra** in 56% yield (Table 2). It is noteworthy that bis-sulfoximine **3ra** could be utilized as a ligand for transition-metal catalysis, as evidenced by its structural similarity to known bis-sulfoxide ligands[11]. *N*-Heteroaryl-diheteroaryl sulfoximines are a class of compounds that have been challenging to prepare, as judged by the lack of examples in the literature. Our method enabled the preparation of one such member of this class in 42% yield (**3sm**, Table 2).

**Mechanistic studies**. We performed a series of experiments to gain insight into the reaction mechanism. Following our hypothesis of photoredox C–N coupling, the impact of light on the reaction was explored. When the standard reaction of **1a** and **2a** was performed in the dark, **3aa** was only obtained in 8% yield after 48 h (Fig. 2a), indicating the pivotal role of ambient light on the conversion. To test whether the reaction proceeds via a radical pathway or a Cu–H intermediate, addition of one equiv 2,2,6,6-tetramethylpiperidine-1-oxyl (TEMPO) to the reaction of **1a** with **2a** was performed. TEMPO–H (**4**) was observed in 10% yield, along with 16% **3aa** (Fig. 2b). Likewise, the introduction of

**Table 1 Optimization of copper-catalyzed photoredox Chan-Lam coupling of 1a with 2a.**

| Entry | catalyst/mol % | Solvent | Conc./M | Assay yield[a]/% |
|---|---|---|---|---|
| 1 | Cu(OAc)$_2$·H$_2$O/20 | MeOH | 0.3 | 25 |
| 2 | CuCl/20 | MeOH | 0.3 | 5 |
| 3 | CuF$_2$/20 | MeOH | 0.3 | 30 |
| 4 | Cu(O$_2$CCF$_3$)$_2$·H$_2$O/20 | MeOH | 0.3 | 36 |
| 5 | Cu(O$_2$CCF$_3$)$_2$·H$_2$O/20 | EtOH | 0.3 | 90 |
| 6 | Cu(O$_2$CCF$_3$)$_2$·H$_2$O/20 | iPrOH | 0.3 | 49 |
| 7 | Cu(O$_2$CCF$_3$)$_2$·H$_2$O/20 | tBuOH | 0.3 | 0 |
| 8 | Cu(O$_2$CCF$_3$)$_2$·H$_2$O/10 | EtOH | 0.3 | 90 |
| 9 | Cu(O$_2$CCF$_3$)$_2$·H$_2$O/5 | EtOH | 0.3 | 45 |
| 10[b] | Cu(O$_2$CCF$_3$)$_2$·H$_2$O/10 | EtOH | 0.3 | 90 |
| 11[c] | Cu(O$_2$CCF$_3$)$_2$·H$_2$O/10 | EtOH | 0.3 | 75 |
| 12[b] | Cu(O$_2$CCF$_3$)$_2$·H$_2$O/10 | EtOH | 1.5 | 95(94[d]) |
| 13[b] | Cu(O$_2$CCF$_3$)$_2$·H$_2$O/10 | EtOH | 2.0 | 85 |
| 14[b] | - | EtOH | 1.5 | 0 |
| 15[e] | Cu(O$_2$CCF$_3$)$_2$·H$_2$O/10 | EtOH | 1.5 | 17(20[f]) |
| 16[g] | Cu(O$_2$CCF$_3$)$_2$·H$_2$O/10 | EtOH | 1.5 | 66 |

General conditions: unless otherwise stated, reactions were carried out with **1a** (0.3 mmol) and **2a** (2.3 equiv) at room temperature under argon for 48 h.
[a]Assay yields determined by $^1$H NMR using 0.1 mmol CH$_2$Br$_2$ (7.0 μL) as internal standard.
[b]**2a** (2.0 equiv).
[c]**2a** (1.5 equiv).
[d]Isolated yield.
[e]Phenyl boronic acid pinacol ester employed.
[f]B(OH)$_3$ (3.0 equiv) as an additive.
[g]Potassium phenyl trifluoroborate (3.0 equiv) employed for 72 h.

1.0 equiv of 5,5-dimethyl-1-pyrroline N-oxide (DMPO) under otherwise standard reaction conditions with **1a** and **2a** resulted in the isolation of radical trapped complex **5**. No **3aa** was observed in the presence of DMPO (Fig. 2c). The HAT product **5** was confirmed by both electron paramagnetic resonance (Supplementary Fig. 1) and high-resolution mass spectrometry (HRMS) analysis (Supplementary Fig. 2).

As hydrogen radicals and copper hydrides have been proposed in copper-catalyzed hydrogen evolution reactions[41–45], we hypothesized that the byproduct of our reaction might be dihydrogen. Examination of the reaction headspace by gas chromatography (GC) confirmed the formation of hydrogen gas (Supplementary Fig. 3).

A light on/off experiment was performed (Fig. 2d) to further explore the impact of irradiation with ambient fluorescent lights on reaction conversion. This experiment indicates that the copper-catalyzed Chan-Lam coupling indeed is a photoredox-catalyzed process rather than a radical-chain pathway. Upon plotting the data, we were surprised to find an unusual increase in the rate of conversion during the second "light-on" period (24–36 h, Fig. 2d), which inspired us to further probe this phenomenon.

To explore the reaction time course, the copper-catalyzed photoredox Chan-Lam coupling was followed under constant ambient light exposure (Fig. 3a). At the initial stage of the reaction (0–12 h), the conversion to **3aa** increased slowly and reached only 11%. The reaction entered a dormant stage (12–32 h), during which very little additional **3aa** was generated (conversion increased from 11% to 15%). Starting after 32 h, the reaction entered an acceleration stage, and conversion to **3aa** jumped to 89% within 10 h. With the consumption of reactants **1a** and **2a** nearly complete, production of **3aa** slowed, eventually

reaching 95% after 48 h. We hypothesized that the acceleration phase might be owing to photoredox autocatalysis[46–48], wherein the product of the reaction becomes the catalyst. This possibility was subsequently explored.

As sulfoximines have been widely utilized as ligands for transition-metal catalysis[4,10,11], we speculated that the autocatalysis might arise from the in situ formation of a copper complex ligated by the N-arylated sulfoximine product. To test this hypothesis, we spiked the reaction mixture of **1a** and **2a** with product **3aa** and monitored the conversion under otherwise standard conditions. When the reaction mixtures were spiked with 20 mol % **3aa** and the reaction time course followed, **1a** and **2a** were steadily consumed over 40 h and **3aa** was produced (Fig. 3b). Initiation of the reaction in the presence of 33 mol % **3aa** led to increased acceleration in the conversion of **1a** and **2a** to the product in under 10 h. These results are consistent with autocatalysis by **3aa**.

We next examined the reaction mixtures by HRMS[31,49] with the goal of detecting intermediates in the transformation of **1a** and **2a** to **3aa**. During the first stage of the reaction (0–12 h), copper(I) ligated with two NH-sulfoximines **1a** (complex **A**) was the major copper species detected by HRMS (Supplementary Fig. 4a–b). As the reaction progressed under argon and **3aa** was generated, ligand exchange was observed by HRMS to generate a copper complex bound to two equiv **3aa** (complex **B**) (Supplementary Fig. 4c–g).

To probe the role of **3aa** as ligand for the copper-photocatalyst in the photoredox Chan-Lam coupling, arylation of methyl phenyl NH-sulfoximine (**6**) with phenylboronic acid (**2a**) was investigated in the presence of 20 mol% **3aa** (Supplementary Table 1). Under otherwise standard conditions, 80% yield of N-arylated product **7** was obtained. In sharp contrast, the yield of **7**

**Table 2 Substrate scope of copper-catalyzed photoredox Chan-Lam coupling.**

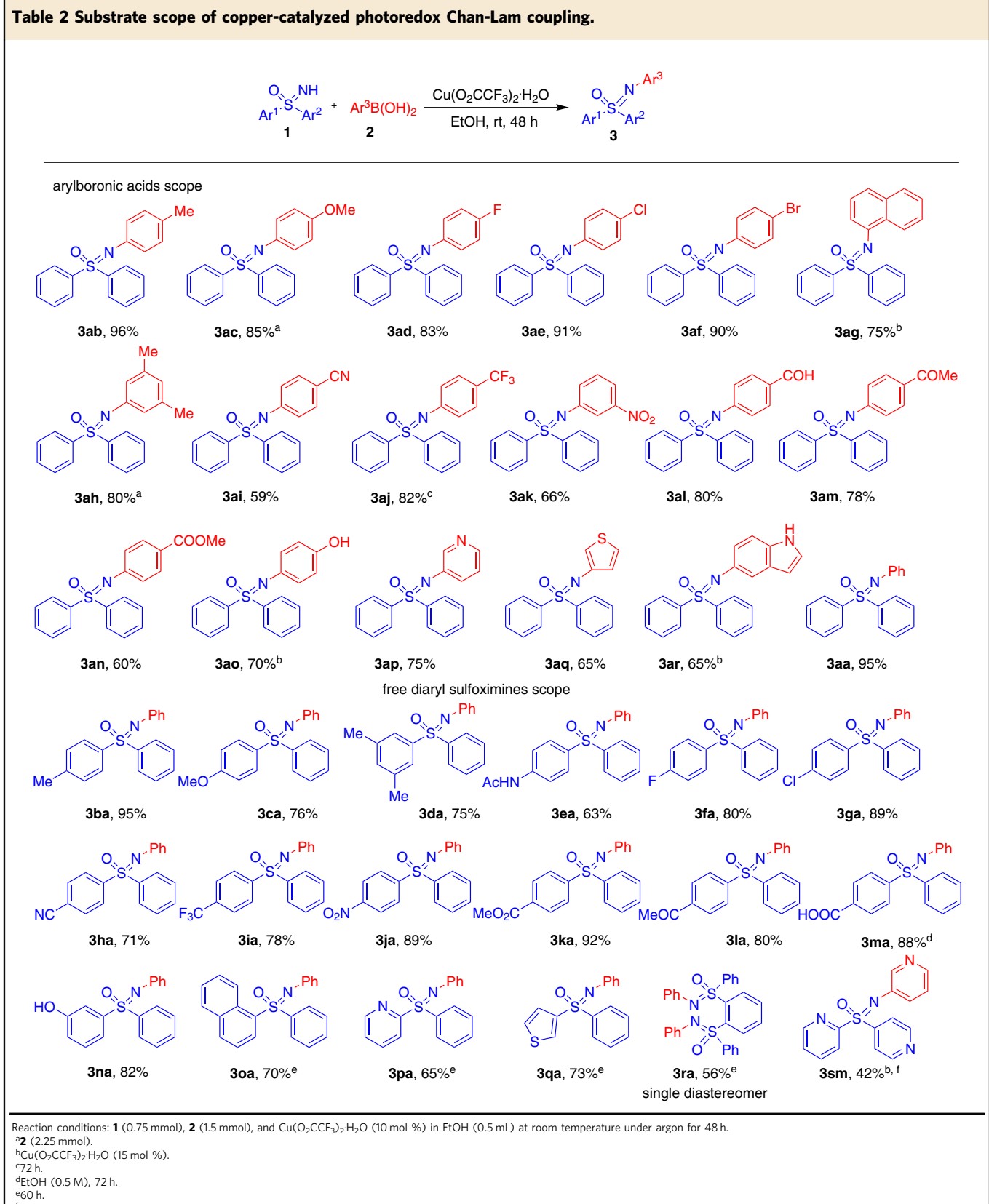

arylboronic acids scope

3ab, 96% — 3ac, 85%[a] — 3ad, 83% — 3ae, 91% — 3af, 90% — 3ag, 75%[b]

3ah, 80%[a] — 3ai, 59% — 3aj, 82%[c] — 3ak, 66% — 3al, 80% — 3am, 78%

3an, 60% — 3ao, 70%[b] — 3ap, 75% — 3aq, 65% — 3ar, 65%[b] — 3aa, 95%

free diaryl sulfoximines scope

3ba, 95% — 3ca, 76% — 3da, 75% — 3ea, 63% — 3fa, 80% — 3ga, 89%

3ha, 71% — 3ia, 78% — 3ja, 89% — 3ka, 92% — 3la, 80% — 3ma, 88%[d]

3na, 82% — 3oa, 70%[e] — 3pa, 65%[e] — 3qa, 73%[e] — 3ra, 56%[e] — 3sm, 42%[b, f]

single diastereomer

Reaction conditions: **1** (0.75 mmol), **2** (1.5 mmol), and Cu(O$_2$CCF$_3$)$_2$·H$_2$O (10 mol %) in EtOH (0.5 mL) at room temperature under argon for 48 h.
[a]**2** (2.25 mmol).
[b]Cu(O$_2$CCF$_3$)$_2$·H$_2$O (15 mol %).
[c]72 h.
[d]EtOH (0.5 M), 72 h.
[e]60 h.
[f]EtOH (0.3 M).

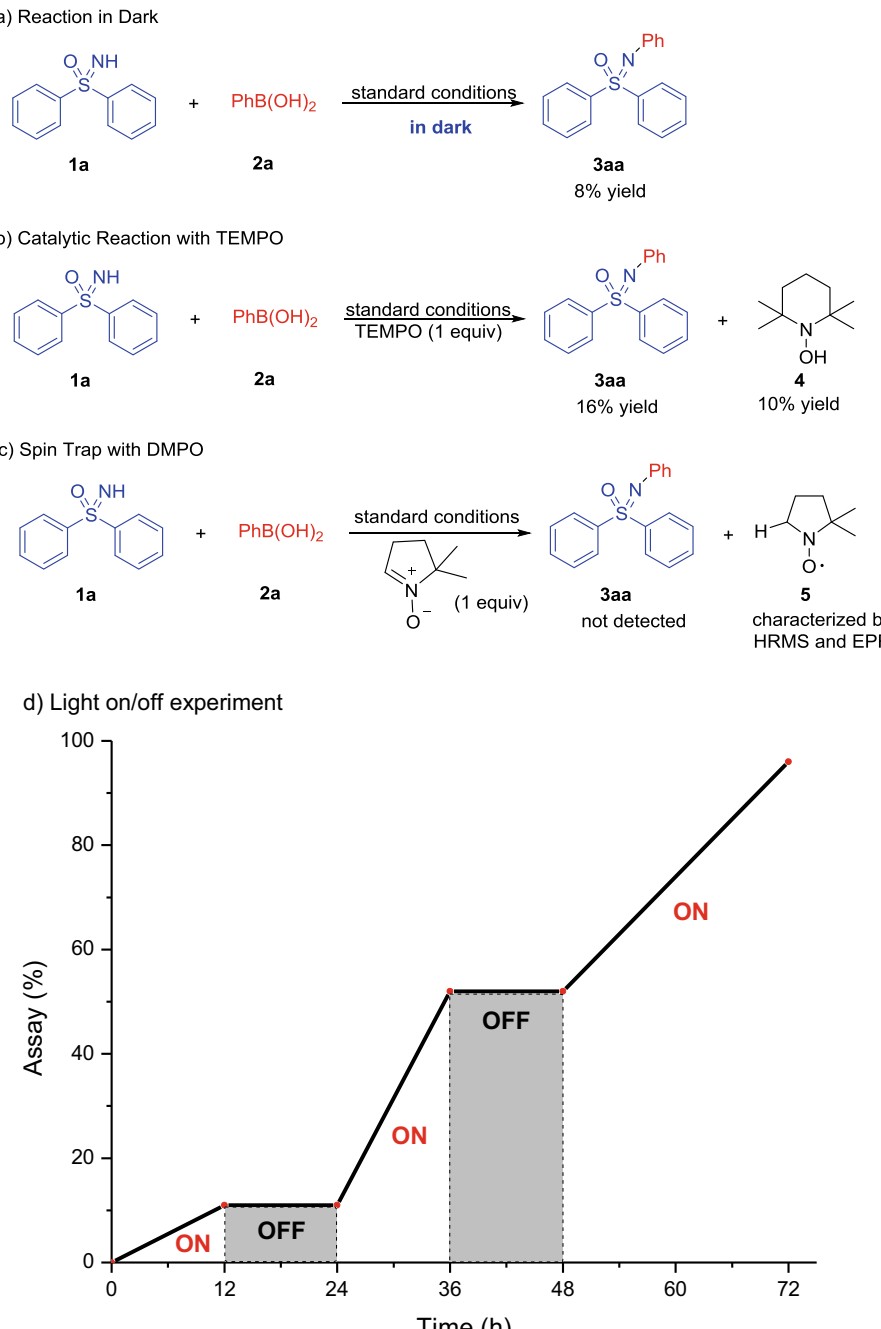

**Fig. 2 Mechanistic probes. a** Examination of the role of ambient light. **b** Probing the radical pathway of copper-catalyzed Chan-Lam coupling. **c** Spin trapping experiments. **d** Light on/off experiment with copper-catalyzed Chan-Lam coupling reaction of **1a** and **2a**.

was only 7% if **3aa** was not present (Supplementary Table 1). The marked difference in arylation yields clearly demonstrates that only *N*-arylated, and not the *NH*-diaryl sulfoximines-ligated copper complexes serve as competent catalyst for this photoredox Chan-Lam coupling.

Spectroscopic studies on the catalyst species were next performed. UV−Vis spectra of **1a**, **2a**, **3aa**, and Cu(NCCH$_3$)$_4$PF$_6$ were recorded (Fig. 4a). As expected, these species do not exhibit appreciable absorption above 245 nm. In contrast, independently synthesized copper(I) hexafluorophosphate complexes [Cu(N(H)OSPh$_2$)$_2$][PF$_6$] (**A'**) and [Cu(N(Ph)OSPh$_2$)$_2$][PF$_6$] **B'** (Supplementary Fig. 5) each exhibited a strong absorption that tails into the visible (Fig. 4a). In addition, emission and excitation spectra of **A'** and **B'** confirmed that they exhibit very similar accessible

excitations in the near-ultraviolet region (Supplementary Fig. 6). When the reaction mixture of **1a** and **2a** (0.5 M) was monitored, the max absorption was ~782 nm, which is indicative of a copper (II) species as the catalyst resting state (Fig. 4a).

A series of Stern−Volmer experiments of complex **B'** utilizing **1a**, EtOH, or **2a** as additive were performed (Supplementary Fig. 7b). The results clearly show that sulfoximine **1a** could effectively quench the excited copper(I) complex **B'** (Fig. 4b). In sharp contrast, solvent ethanol and **2a** exerted negligible quenching effect on excited **B'** (Supplementary Fig. 7b). Similar quenching studies with complex **A'** were also conducted, but surprisingly excited **A'** was not quenched by **1a** (Fig. 4b), EtOH, or **2a** (Supplementary Fig. 7a). This unexpected result excludes the possibility of **A'** acting as a competent photocatalyst in this

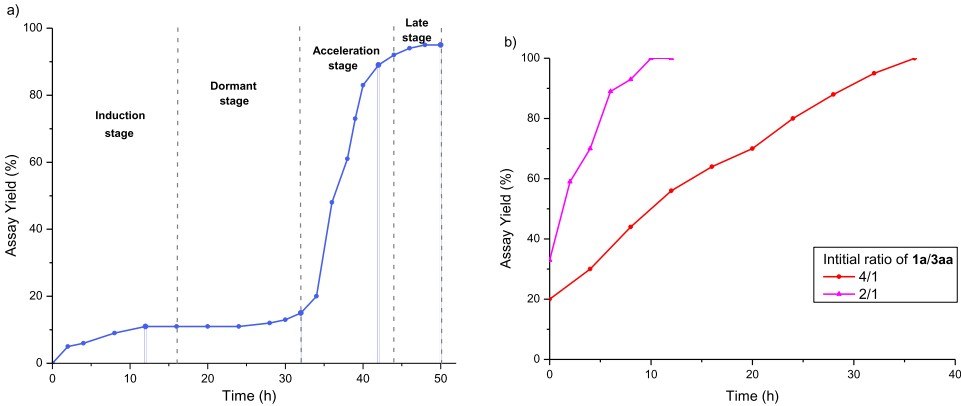

**Fig. 3 Reaction profiles of copper-catalyzed photoredox Chan-Lam couplings. a** Standard conditions. **b** Initiation in the presence of **3aa**.

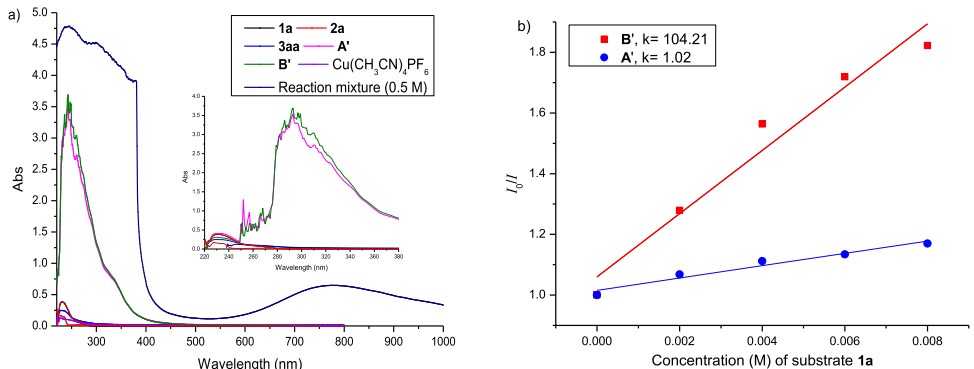

**Fig. 4 Spectroscopic studies. a** UV-Vis spectra of **1a**, **2a**, **3aa**, **A′**, **B′** and Cu(NCCH$_3$)$_4$PF$_6$ in DCM (5 × 10$^{-5}$ M) and reaction mixture in EtOH (**1a**, 0.5 M). **b**, Stern–Volmer quenching efficiency of **1a** with **A′** and **B′** (5 × 10$^{-5}$ M in DCM, $\lambda_{ex}$ = 344 nm).

Chan-Lam coupling. Simple copper(I) species Cu(NCCH$_3$)$_4$PF$_6$ did not exhibit emission quenching when treated with the above additives either (Supplementary Fig. 7c). Taken together, these data allow us to propose that the autocatalysis phenomenon is owing to ligation of the *N*-arylated sulfoximines **3** to copper(I) species.

Based on the results above and reports in the literature, a preliminary reaction pathway is proposed (Fig. 5). During the initial stoichiometric stage, a copper(II)-mediated C–N coupling occurs between free sulfoximine (**1**) and arylboronic acid (**2**) to give a small amount of *N*-aryl sulfoximine (**3**) (induction period, Fig. 3a). Here, a transmetallation between [Cu$^{II}$] (**C**) and arylboronic acid (**2**) proceeds to yield Ar$^3$–Cu(II) (**D**) (step *i*). Following coordination of *NH*-sulfoximine (**1**) and deprotonation to form **E** (step *ii*), **E** undergoes a disproportionation to produce Cu(III) species **F** and a Cu(I) species (step *iii*) in a fashion similar to that proposed by the groups of Watson[31,32] and Stahl[33,50]. Intermediate **F** undergoes reductive elimination (step *iv*) to generate product **3** and another equiv. of Cu(I). The stoichiometric reaction converts Cu(II) to Cu(I) and forms a small amount of **3**. A traditional Chan-Lam coupling would involve oxidation of Cu(I) **G** back to Cu(II) **C** (step *v*), but our reaction does not employ added oxidant. The reaction next enters the dormant phase, only slowly generating product **3**.

The next phase of the reaction is the photo- and auto-catalytic phase, which has no precedent in the literature. As supported by Stern−Volmer experiments, the Cu(I) complex of the *NH*-sulfoximine, [Cu$^I$(**1**)$_2$]$^+$ (**C′**), is not a photocatalyst. The *N*-Ph sulfoximine (**3**)-ligated Cu(I) complex, [Cu$^I$(**3**)$_2$]$^+$ (**D′**) can be quenched by **1** and is a potential photocatalyst. Interestingly,

calculations indicate that [Cu$^I$(**3**)$_2$]$^+$ (**D′**) possesses the highest energy (13.9 kcal/mol) among **C′**, **D′**, and **E′** (Supplementary Fig. 8), which indicates it is the least stable Cu(I) species, and therefore, is less likely to serve as the actual photocatalyst in the transformation. In contrast, mixed species [Cu$^I$(**1**)(**3**)]$^+$ (**E′**) (4.4 kcal/mol, Supplementary Fig. 8), which can be generated from **C′** or **D′** via a ligand exchange process at room temperature, is believed to facilitate the following photocatalytic process. The catalytic photoredox Chan-Lam coupling cycle is proposed to commence with the excitement of **E′** by ambient light (step *i′*) to afford [Cu$^I$(**1**)(**3**)]$^{+*}$ (**F′**), followed by metal-to-ligand charge transfer (MLCT, step *ii′*) to yield [Cu$^{II}$(**1**)(**3**)]$^+$ (**F′**), featuring a radical anion *NH*-sulfoximine ligand. Meanwhile, the solvent EtOH coordinates to the Cu center to afford **G′**. Cu(II)-facilitated homolysis of O–H bond of EtOH occurs to give a formal Cu(III) species **H′** (step *iii′*)[51]. The concomitant hydrogen radical, either free or coordinated by ethoxide, attacks the hydrogen in *NH*-sulfoximine ligand of **H′** to release dihydrogen gas, and generate a formal Cu(III) species **I′** simultaneously (step *iv′*). Transmetallation of Ar$^3$–B(OH)$_2$ (**2**) (step *v′*) with copper(III) (**I′**) produces the Cu$^{III}$–Ar$^3$ complex (**J′**), that undergoes reductive elimination (RE)[51] and ligand exchange (step vi′) with **1** to generate **3** and close the catalytic cycle. The generated *N*-aryl sulfoximine (**3**) can bind copper(I), increasing the amount of active copper catalyst and giving rise to autocatalysis. The existence of hydrogen radicals was supported by the generation of TEMPO–H (Fig. 2b) and the spin trapped product (Fig. 2c). Consistent with the proposed mechanism, hydrogen gas was detected in the reaction headspace by GC (Supplementary Fig. 3).

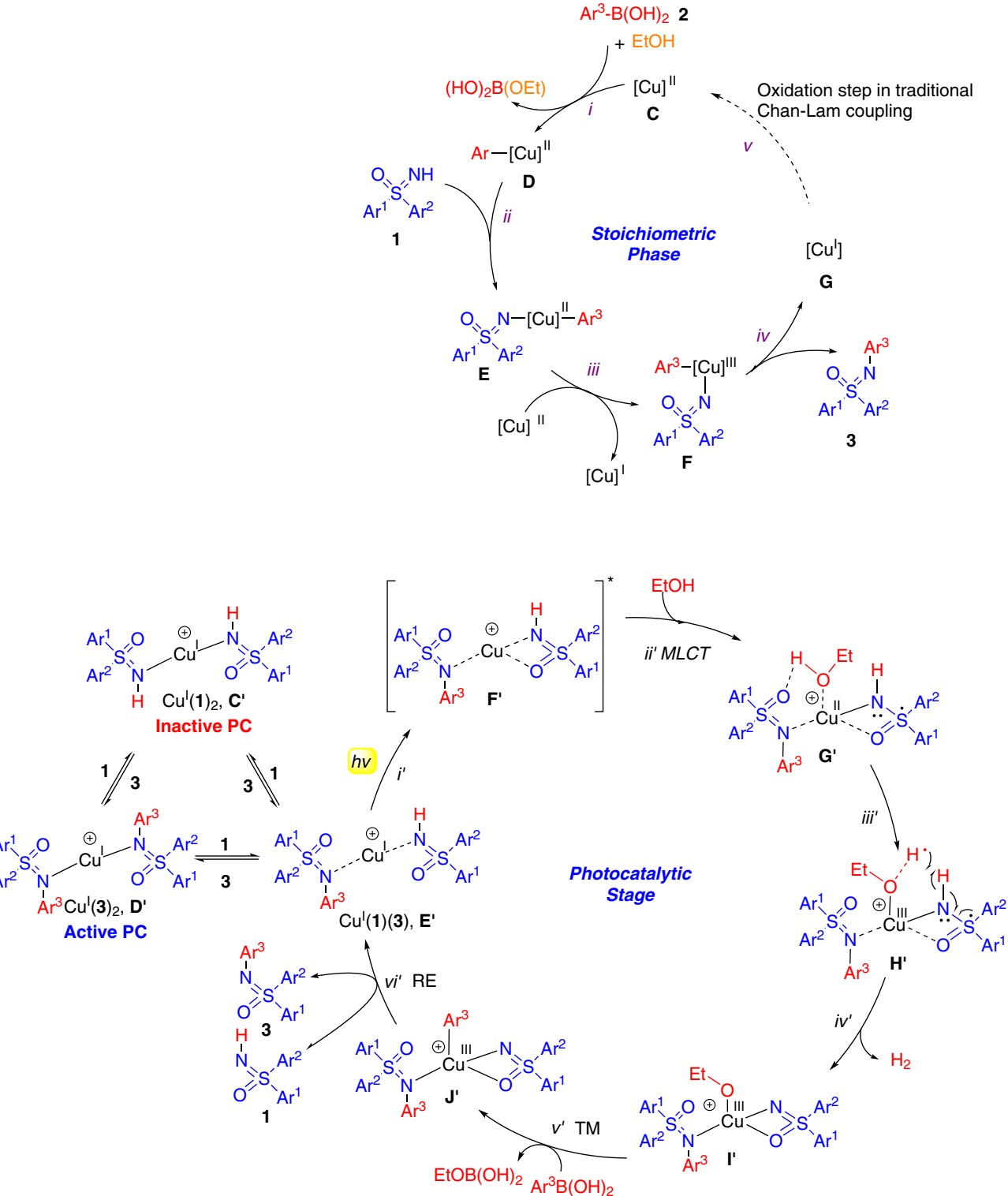

**Fig. 5 Proposed mechanism of copper-catalyzed photoredox Chan-Lam coupling.** The reaction was initiated by a stoichiometric classical oxidative Chan-Lam coupling, and then entered the photocatalytic stage after *N*-arylted product **3** and *NH*-sulfoximine **1** binding to Cu(I) together to form the active photocatalyst **E'**.

**Computational studies**. To understand the nature of the photocatalyst and how hydrogen gas forms, a density-functional theory study of the transformation was initiated (see Supplementary Data 1 for details). According to the proposed mechanism (Fig. 5), the photocatalyst formed in the stoichiometric phase undergoes a MLCT after exposure to light. Alcohol can coordinate to the copper and together with the sulfoximine loses a hydrogen atom to form hydrogen gas. The rest of the cycle is well-precedented for Chan-Lam couplings. The thermodynamics and pathways of key steps were calculated with

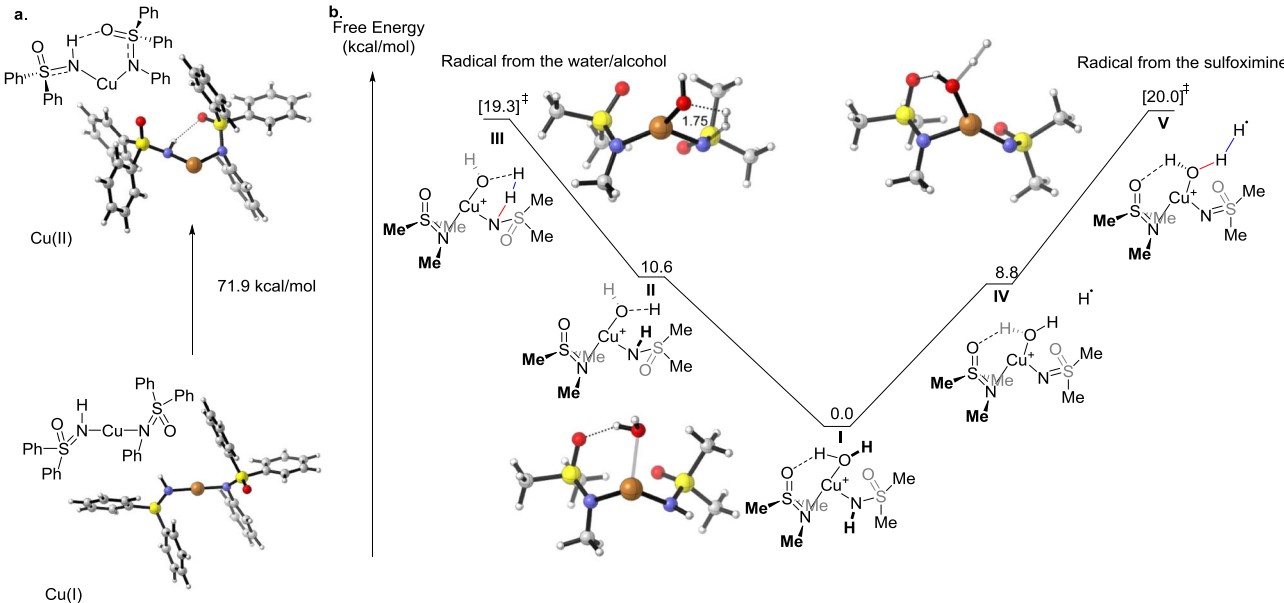

**Fig. 6 Computed pathways. a** Structures and energy difference of copper(I) and copper(II) photocatalysts. **b** Energy profiles for the formation of hydrogen gas through radical mechanisms. Free energies computed using B3LYP/6-31 G(d), Cu:SDD.

B3LYP/6-31 G(d), Cu:SDD[52–54] using Gaussian 16[55], and graphics were made in Cylview[56].

For both the copper (I) and copper (II) species, the most stable species had two molecules of substrate **1** coordinated and the least stable species had two molecules of the product **3** coordinated (Supplementary Fig. 8). In general, the substrate coordinated species were stabilized by hydrogen bonding between the NH of one sulfoximine and the S=O of another. The lesser stability of the 2:1 product:Cu adducts indicates that product exchange would be favorable consistent with turnover occurring. Although, not isolated in preparations, the 1:1:1 substrate: product: Cu complexes were found to be of intermediate stability and would be accessible under the reaction conditions (Supplementary Fig. 8). These complexes were stabilized by similar hydrogen bonds as found in the substrate adducts. It appears likely that the 1:1:1 substrate: product: Cu complex is the active catalyst, but it is unclear if it forms before or after photoexcitation and MLCT. However, the energy needed to excite the copper(I) hetero-complex was lower than either of the homo-complexes (Supplementary Fig. 8 and 6a) and corresponds to light around 400 nm. For these reasons, this mixed species was chosen as a starting point for the pathway calculations.

For the computational analysis of the different pathways, methyl groups were used in place of aryl groups and water was used in place of ethanol in order to reduce computation times. Three different mechanisms for the formation of hydrogen gas were considered: (1) a copper hydride species, (2) H• arising from the alcohol, and (3) H• arising from the sulfoximine. The copper hydride pathway has a barrier of 27.6 kcal/mol to forming the copper hydride from the starting sulfoximine (Supplementary Fig. 9) which is inconsistent with reaction at room temperature. Experimental results described above also indicate that TEMPO inhibits the reaction, which is further inconsistent with a copper hydride pathway[57].

Examination of the H• pathways revealed two possibilities (Fig. 6b). The mechanism on the left (**I→II→III**) forms hydrogen gas through a H• from the water. The hydrogen dissociates from the water to form an oxygen-associated H• in **II** (O–H bond length = 2.1 Å). This oxygen-associated H• then abstracts the hydrogen on the nitrogen of the sulfoximine via transition state

**III** to form hydrogen gas. The mechanism on the right to forms (**I→IV→V**) forms hydrogen gas through a H• from the sulfoximine. The H• first dissociates from the sulfoximine to give **IV**. This H• then abstracts a hydrogen from the water via transition state **V**. Although transition state **III** is slightly lower in energy than transition state **V**, both pathways are plausible at room temperature.

In summary, an photoredox copper-catalyzed dehydrogenative Chan-Lam coupling of free diaryl sulfoximines and arylboronic acids has been introduced. An array of functional groups, including heteroarylboronic acids and free heteroaryl sulfoximines are viable. Mechanistic investigations reveal a unique autocatalysis process. The ligation of *N*-arylated sulfoximines to copper species enables its capability as photocatalyst, which efficiently facilitates the C–N coupling process in the absence of external oxidant. Understanding the nature of the autocatalysis raises the possibility to tailor copper catalysts with specific ligands to facilitate photoexcitation. The protocol described herein represents an appealing alternative strategy to the classic oxidative C–N coupling process, allowing the expansion of substrate generality as well as the elimination of byproducts from catalyst oxidation. The concept of sacrificial oxidant-free photoredox Chan-Lam coupling provides a basis for the design of systems to allow the formation of C–N bonds in other contexts.

## Methods

General procedure for catalysis: to an oven-dried microwave vial equipped with a stir bar was added sulfoximine **1** (0.75 mmol, 1.0 equiv), boronic acid **2** (1.5 mmol, 2.0 equiv), Cu(O₂CCF₃)₂·H₂O (21.7 mg, 0.075 mmol, 10 mol %) under an argon atmosphere in a dry box. The vial was capped with a septum and removed from the dry box. EtOH (0.5 mL) was added into the reaction vial via syringe, and the reaction solution was stirred at room temperature under argon with ambient light for 48 h. Upon completion of the reaction, the vial was opened to air, and the reaction mixture was passed through a short pad of silica gel. The pad was then rinsed with 100:1 dichloromethane:methanol (20.0 mL). The solvent was removed under reduced pressure. The residue was purified by flash chromatography to afford the purified product.

## Data availability

Experimental procedure and characterization data of new compounds are available within Supplementary Information. Any further relevant data are available from the authors upon reasonable request.

# ARTICLE

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

## Acknowledgements

T.J. thanks Shenzhen Nobel Prize Scientists Laboratory Project (C17783101), the Science and Technology Innovation Commission of Shenzhen Municipality (JCYJ20180302180256215), and Guangdong Provincial Key Laboratory of Catalysis (2020B121201002) to provide financial support. SUSTech is gratefully acknowledged for providing startup funds to T.J. (Y01216129). We are also very grateful to professor Lele Duan and Dr. Baihua Long (SUSTech) for the assistance of GC, professor Wei Lu and Xiaobao Zhang (SUSTech) for assistance of Stern−Volmer experiments, and Dr. Yang Yu (SUSTech) for HRMS. P.J.W thanks the US National Science Foundation (CHE-1902509) for financial support. M.C.K. thanks the NIH (GM131902) for financial support and XSEDE (TG-CHE120052) for computational support. This paper is dedicated to the 10th anniversary of Department of Chemistry, SUSTech.

## Author contributions

T.J. designed and supervised the project. C.W., H.Z., T.L., and T.M. performed the experiments. M.C.K. directed the computational study. L.A.W. carried out the computational study. P.J.W. suggested mechanistic experiments. H.Z., T.L., and Q.L. analyzed the results. T.J., P.J.W., and M.C.K. wrote the paper. C.W., H.Z., and L.A.W. contributed equally.

## Competing interests

The authors declare no competing interests.
