## [Peer Review File · Nature Communications]

REVIEWER COMMENTS

Reviewer #1 (Remarks to the Author):

Jia and coworkers described a copper-catalyzed photoredox dehydrogenative Chan-Lam coupling reaction to synthesize N-aryl sulfoximine derivatives. The key feature is that the reaction might go through a photoredox autocatalytic process in the absence of any oxidants. There are a few examples using diaryl sulfoximines as substrates, and their high yields showed the versatility of this reaction in previous work (Synthesis 2019, 51, A; Adv. Synth. Catal. 2012, 354, 986; J. Org. Chem. 2018, 83, 11369; Org. Lett. 2019, 21, 2740; Tetrahedron Lett. 2016, 57, 2372; Tetrahedron Lett. 2020, 61, 152079). Moreover, copper-catalyzed N-arylations of sulfoximines with arylboronic acids under mild conditions have been developed by Bolm (Org. Lett. 2005, 7, 2667; Adv. Synth. Catal. 2018, 360, 1088) and Kandasamy (Synthesis 2019, 51, A). Considering that the reaction might undergo a new mechanism, we hope that the author could resubmit this manuscript for review after completing the following comments.

(1) In line 55, König and Wimmer's work have not been cited.

(2) For the photoredox N-arylation of NH-sulfoximines, the following references should be added: Org. Lett. 2019, 21, 2740. The references (Org. Lett. 2005, 7, 2667; Synthesis 2019, 51, A) on copper-catalyzed N-arylation of sulfoximines with arylboronic acids also need to be cited.

(3) In Table 1, the substrate scope showed that the reactions of electron-neutral, electron-donating and electron-deficient phenylboronic acid substrates proceeded smoothly. Have the authors studied more electron-deficient phenylboronic acid, such as -CF₃, -NO₂ and -CN substituted phenylboronic acid?

(4) In Table 1, only arylboronic acids were employed in this work. How about other aryl sources, such as phenyltrifluoroborate, vinylboronic acid and phenylboronic acid pinacol ester?

(5) In Table 1, for the functional group tolerance, how about diaryl sulfoximines with hydroxyl, ester, amide, cyanide or nitro substituents?

(6) The proposed mechanism sounds reasonable. For the highlight of this manuscript is the new reaction mechanism, the authors should provide more experimental evidences for the mechanism, such as DFT calculations or ST/EPR experiments.

Reviewer #2 (Remarks to the Author):

Transition metal catalyzed N-arylation of sulfoximine is well-explored reaction in the literature, have been achieved with various aryl donors including aryl halides, arylboronic acids, etc (J. Sulfur Chem. 2018, 39, 674). Bolm group has introduced the copper catalyzed N-arylation of sulfoximines with simple arylboronic acids in 2005 (Org. Lett. 2005, 7, 2667), while Kandasamy et al. have explored the N-arylation of sulfoximines with simple as well as sterically hindered arylboronic acids at room temperature with large number of sulfoximine examples including diphenyl sulfoximine in high yields (Synthesis, 2019, 51, 2171-2182).

In the current paper, the authors (Jia et al) have mainly focused on the copper catalyzed N-arylation of di-arylsulfoximines stating that they are quite unreactive due to less nucleophilicity. To some extent, I can agree to the fact that many groups did not focus on the N-arylation of di-arylsulfoximines. On the other hand, it was not so important to be focused. The positive point of the current manuscript is that they have studied the mechanism of the reaction and identified autocatalytic photoredox progress.

However, having plenty reports on N-arylation of sulfoximine in the literature, I would like to state that this methodology has lack of novelty and do not deserve to be published in Nature Communication.

Reviewer #3 (Remarks to the Author):

This paper is an interesting report on a catalytic photoredox Chan-Lam coupling of diaryl sulfoximes with arylboronic acids to give N-arylated products.

As the reaction is copper catalyzed, much of the discussion is devoted to the role of coppers' different presumed oxidation states (+I, +II or +III) in the complexes involved.

Important is the observation of dihydrogen evolution, which indicates that either hydrogen atoms or hydride must be involved in the reaction. The authors conclude that copper-hydride forms (box B, Fig. 5) as a result of the prior reduction of the NH species by an excited state complex and that eventually ethanol is a proton source (box A, Fig. 5), as it is obviously assumed to be a stronger acid than the (unnamed) boronic acid species in box A.

Another important observation is that the reaction does hardly occur in the dark. Remarkably, there is a 15 h long "dormant" phase with only negligible product generation. The yield S-shaped curve after the "dormant" stage indicates an autocatalytic process, as the authors state. This was also proven by initial addition of 3aa to the reactants (even though that experiment described in Table S2 does not prove that only N-arylated species are involved in photocatalytic complexes).

This work reports very important observations but some inconsistencies are to be removed and some improvements should be made before the paper should be published.

1.) I doubt that EtO⁻ is a weaker base than CF₃COO⁻ (Box A in Fig. 5)! Why do the authors believe that ?

2.) Does the hydrogen evolution really sets in only after more than 30 h reaction time (Fig. 3) as Fig. 5 suggests ?

3.) The species in the "photocatalytic stage" cycle should all be numbered (not only the steps), similar to the so-called "stoichiometric phase" cycle above. The stoichiometries must be checked, the total charges of all complexes given, ligands must be specified in the caption of Fig. 5 or the figure itself (e.g. L = ? n = ?).

It is not always clear why species involved in the "stoichiometric phase" look so different as those in the second (catalytic) cycle below, after the reduction of hydrogen to hydride by the excited state Cu(I) complex therein. E.g. in "E", only N is shown to coordinate to copper, while in the cycle below, both N and O coordinate (if L is anionic, what is then index n? if L is neutral, is the charge correct?). The formal Cu(III) species is charged in the second cycle but uncharged in the first etc. Without exactly specifying all items, it is difficult to follow the proposal. Step iv in the first cycle has the acetate appearing as a ligand. Where does it come from as L is apparently not acetate (line 242)?

4.) What are the complex formation constants for A' and B' ? Is the equilibrium shown to the left in the second cycle in Fig 5 between Cu(I) (1)₂ and Cu(I) (3)₂ (a consistent numbering is needed again!) possible at all ? How about having a Cu species with mixed ligands ? Is A' indeed transforming through ligand exchange into B' ? And how fast is this process?

5.) How to explain the induction stage and the subsequent dormant phase? The authors assume that ligands 1a (in contrast to 3aa) do not form photocatalytic copper complexes based on a control experiment (Table S2) and their Stern-Volmer plots of fluorescence quenching.

However, the Stern-Volmer constant for complex A' is not zero and the complex definitely also absorbs visible light (Fig. 4).

I do not understand why the absence of an external oxidant should result in the curiously long "dormant phase" under irradiation. Wouldn't it be more plausible to assume that some transformation of more active copper complexes has to occur first, before the second self-accelerating autocatalytic phase (this time by product 3aa rather than reactant 1a) sets in ?

The photocatalysis could therefore also be by complexes formed from the NH rather than NR species alone, i.e. might involve also the reactant 1a (!) and not only the product 3aa as ligands. This would also explain the self-decelerating "induction stage" (Fig.3a). See for a recent paper which should be cited and discusses an autocatalysis by the reactant: ChemPhysChem 2020, 21, 1775.

6.) I also wondered that the authors did not try to do an experiment with an UV lamp to see whether the rates and yields increase, seeing that absorption in the visible region is very small. It is really difficult to understand this, especially as alone the Cu(II) complex (Fig. 4 and 5) is apparently absorbing visible light to a significant extent. Does the reaction e.g. still proceed when the absorption of visible light is prevented under UV light irradiation (by use of appropriate filters)? Such an experiment would be really helpful.

7.) Shouldn't the actual resting state be the Cu(I) ground state, rather than a Cu(II) species resulting from the redox reaction? The reality of Cu(III) oxidation state has been very recently questioned, that paper needs to be cited and discussed and the proposed mechanism perhaps revised, see: JACS 2019, 141, 18508. This could also affect the disproportionation of Cu (II) into Cu(I) and Cu(III), which the authors postulate in Fig. 5.

Reviewer #1 (Remarks to the Author):

Jia and coworkers described a copper-catalyzed photoredox dehydrogenative Chan-Lam coupling reaction to synthesize N-aryl sulfoximine derivatives. The key feature is that the reaction might go through a photoredox autocatalytic process in the absence of any oxidants. There are a few examples using diaryl sulfoximines as substrates, and their high yields showed the versatility of this reaction in previous work (*Synthesis* 2019, 51, A; *Adv. Synth. Catal.* 2012, 354, 986; *J. Org. Chem.* 2018, 83, 11369; *Org. Lett.* 2019, 21, 2740; *Tetrahedron Lett.* 2016, 57, 2372; *Tetrahedron Lett.* 2020, 61, 152079). Moreover, copper-catalyzed N-arylations of sulfoximines with arylboronic acids under mild conditions have been developed by Bolm (*Org. Lett.* 2005, 7, 2667; *Adv. Synth. Catal.* 2018, 360, 1088) and Kandasamy (*Synthesis* 2019, 51, A). Considering that the reaction might undergo a new mechanism, we hope that the author could resubmit this manuscript for review after completing the following comments.

(1) In line 55, König and Wimmer's work have not been cited.

Our reply: We really appreciate Reviewer 1's careful reading, and have added two papers (*Adv. Synth. Catal.* **360**, 3277-3285 (2018); *Org. Lett.* **21**, 2740-2744 (2019)) as Reference 22 and 23 in the revised manuscript.

(2) For the photoredox N-arylation of NH-sulfoximines, the following references should be added: *Org. Lett.* 2019, 21, 2740. The references (*Org. Lett.* 2005, 7, 2667; *Synthesis* 2019, 51, A) on copper-catalyzed N-arylation of sulfoximines with arylboronic acids also need to be cited.

Our reply: Following Reviewer 1's suggestion, we have added the citations (*Org. Lett.* **21**, 2740-2744 (2019) as Reference 23; *Org. Lett.* **7**, 2667-2669 (2005) as Reference 14) in the revised manuscript. The paper published by Kandasamy and coworkers (*Synthesis* **51**, 2171-2182 (2019)) was cited in the previous version of the manuscript as Reference 17.

(3) In Table 1, the substrate scope showed that the reactions of electron-neutral, electron-donating and electron-deficient phenylboronic acid substrates proceeded smoothly. Have the authors studied more electron-deficient phenylboronic acid, such as -CF₃, -NO₂ and -CN substituted phenylboronic acid?

Our reply: Following Reviewer 1's suggestion, we have tested and found that arylboronic acids bearing electron-withdrawing 4-CN (**2i**), 4-CF₃ (**2j**), and 3-NO₂ (**2k**) were suitable coupling substrates, delivering the corresponding products (**3ai-3ak**) in 59-82% yields, which have been highlighted by yellow in the revised manuscript.

^a72 h, 3.0 equiv 4-Trifluoromethylphenyl boronic acid (**2j**)

(4) In Table 1, only arylboronic acids were employed in this work. How about other aryl sources, such as phenyltrifluoroborate, vinylboronic acid and phenylboronic acid pinacol ester?

Our reply: Boronic acids are generally employed as coupling partners in Chan-Lam coupling, whereas the utilization of other aryl sources remains challenging. For example, Watson and

coworkers reported the employment of arylboronic acid pinacol esters in oxidative Chan-Lam coupling reactions (*J. Am. Chem. Soc.* **139**, 4769-4779 (2017)), and discovered that the key to success is to add B(OH)₃ as additive to prevent copper catalysis inhibition by pinacol. With this said, using phenylboronic acid pinacol ester instead of phenylboronic acid under the optimal conditions only lead to poor yield of **3aa** (17%). Furthermore, the employment of 3 equiv B(OH)₃ as additive exerted negligible effect on the outcome of the transformation, and only 20% yield of **3aa** was obtained. In contrast, phenyltrifluoroborate turned out to be a compatible partner, and the arylated sulfoximine **3aa** was generated in 66% yield under slightly modified conditions (72 h). Unfortunately, the attempts to arylate free diphenyl sulfoximine **1a** with vinylboronic acid failed, since reactants remained unreactive and no desired product was detected. These findings are described above and below the table (now Table 1).

^a72 h, 3.0 equiv. PhBF₃K
^badded 3.0 equiv. B(OH)₃

(5) In Table 1, for the functional group tolerance, how about diaryl sulfoximines with hydroxyl, ester, amide, cyanide or nitro substituents?

Our reply: The diaryl sulfoximine with amide (**1e**) group has been disclosed in the manuscript, and the desired product **3ea** was produced in 63% yield. We have synthesized all other diaryl sulfoximines bearing hydroxyl (**1n**), ester (**1k**), cyanide (**1h**), nitro (**1j**) and even carboxyl (**1m**) substituents, which were subjected to the copper-catalyzed oxidant-free Chan-Lam coupling reactions. Gratifyingly, all of them performed very well under our catalytic conditions, which have been highlighted by yellow in the revised manuscript.

^aEtOH (0.5 M), 72 h

(6) The proposed mechanism sounds reasonable. For the highlight of this manuscript is the new reaction mechanism, the authors should provide more experimental evidences for the mechanism, such as DFT calculations or ST/EPR experiments.

Our reply: Following Reviewer 1's suggestion, we have conducted the DFT calculations to support the mechanism, which have been disclosed as "Computational Studies" section in detail in the revised manuscript. The calculations provide insight into the possible catalysts that arise from autocatalysis and indicate that the mechanism involves hydrogen atom abstraction steps to generate dihydrogen which is consistent with the experimental findings. The barriers are also consistent with reaction at room temperature as was observed.

Reviewer #2 (Remarks to the Author):

Transition metal catalyzed N-arylation of sulfoximine is well-explored reaction in the literature, have been achieved with various aryl donors including aryl halides, arylboronic acids, etc (J. Sulfur Chem. 2018, 39, 674). Bolm group has introduced the copper catalyzed N-arylation of sulfoximines with simple arylboronic acids in 2005 (Org. Lett. 2005, 7, 2667), while Kandasamy et al. have explored the N-arylation of sulfoximines with simple as well as sterically hindered arylboronic acids at room temperature with large number of sulfoximine examples including diphenyl sulfoximine in high yields (Synthesis, 2019, 51, 2171-2182).

In the current paper, the authors (Jia et al) have mainly focused on the copper catalyzed N-arylation of di-arylsulfoximines stating that they are quite unreactive due to less nucleophilicity. To some extent, I can agree to the fact that many groups did not focus on the N-arylation of di-arylsulfoximines. On the other hand, it was not so important to be focused. The positive point of the current manuscript is that they have studied the mechanism of the reaction and identified autocatalytic photoredox progress.

However, having plenty reports on N-arylation of sulfoximine in the literature, I would like to state that this methodology has lack of novelty and do not deserve to be published in Nature Communication.

Our reply: We would like to thank Reviewer 2 for pointing out the positive point of our work is the discovery of an autocatalytic process. After we carefully read the papers mentioned by Reviewer 2, we respectfully disagree with Reviewer 2's conclusion that "this methodology has lack of novelty". *N*-Aryl diaryl sulfoximines are a class of structural scaffolds of great value in medicinal chemistry as well as drug development, such as EphB4 receptor inhibitor (Figure 1a in manuscript). In sharp contrast to alkyl aryl sulfoximines, the arylation of free diaryl sulfoximines was underexplored. For example, in Bolm's report of copper-catalyzed *N*-arylation of sulfoximines (Org. Lett. 2005, 7, 2667), *none* of the diaryl sulfoximines was employed. In the other paper mentioned by Reviewer 2 (Synthesis, 2019, 51, 2171-2182), the Kandasamy group developed a copper-catalyzed C-N arylation of sulfoximines under mild conditions, but only a single example of free diaryl sulfoximine (diphenyl sulfoximine) was disclosed. As we discussed in the manuscript, a general and practical *N*-arylation of diaryl sulfoximines with broad functional group tolerance remains an unsolved problem.

From the standpoint of mechanism, herein we developed an unprecedented copper-catalyzed external-oxidant-free dehydrogenative photoredox Chan-Lam coupling, which was considered by Reviewer 1 to be "a new mechanism" and "an interesting report" by Reviewer 3. Leveraging MLCT process of copper-based photocatalyst, Cu(I) could be oxidized to Cu(II) in the absence of external oxidant, which prevents the formation of oxidative byproducts. From the kinetic studies, an autocatalytic process was revealed, and the copper species ligated by *N*-arylated sulfoximine/*NH*-sulfoximine was identified as the actual photocatalyst supported by experiments as well as the DFT calculations. We believe, the concept of external-oxidant-free photoredox C-

N arylation will provide basis for the development of new methods complimentary to the classic oxidative coupling, and will find widespread applications in other context. Furthermore, the observation of autocatalysis and identification of the active photocatalysts provided considerable promise for the design of ligands to enable photocatalysis in this and other transformations.

Reviewer #3 (Remarks to the Author):

This paper is an interesting report on a catalytic photoredox Chan-Lam coupling of diaryl sulfoximes with arylboronic acids to give N-arylated products.

As the reaction is copper catalyzed, much of the discussion is devoted to the role of coppers' different presumed oxidation states (+I, +II or +III) in the complexes involved.

Important is the observation of dihydrogen evolution, which indicates that either hydrogen atoms or hydride must be involved in the reaction. The authors conclude that copper-hydride forms (box B, Fig. 5) as a result of the prior reduction of the NH species by an excited state complex and that eventually ethanol is a proton source (box A, Fig. 5), as it is obviously assumed to be a stronger acid than the (unnamed) boronic acid species in box A.

Another important observation is that the reaction does hardly occur in the dark. Remarkably, there is a 15 h long "dormant" phase with only negligible product generation. The yield S-shaped curve after the "dormant" stage indicates an autocatalytic process, as the authors state. This was also proven by initial addition of 3aa to the reactants (even though that experiment described in Table S2 does not prove that only N-arylated species are involved in photocatalytic complexes).

This work reports very important observations but some inconsistencies are to be removed and some improvements should be made before the paper should be published.

1.) I doubt that EtO⁻ is a weaker base than CF₃COO⁻ (Box A in Fig. 5)! Why do the authors believe that ?

Our reply: We agree with Reviewer 3 that ethoxide is a stronger base than trifluoroacetate. Here, we are dealing with a boron alkoxide vs. a trifluoroacetate. The alkoxide will form a stronger interaction with boron than the trifluoroacetate. Additionally, the trifluoroacetate will get syphoned off as it forms the Cu(O₂CCF₃)₂.

2.) Does the hydrogen evolution really sets in only after more than 30 h reaction time (Fig. 3) as Fig. 5 suggests ?

Our reply: According to the results of control reactions, we believe that the hydrogen evolution accompanied the formation of product in the photocatalytic cycle, but does not occur in the dormant stage. The details of control reactions are summarized in Table C1, which is consistent with the proposed mechanism.

Table C1. Control Reactions of H₂ Detection

Entry	1a	2a	Cu(O ₂ CCF ₃)·H ₂ O	EtOH	Time (h)	H ₂
-------	----	----	---	------	----------	----------------

1	1	1	1	✓	12	N.D.
2	1	1	1	✓	12	N.D.
3	✓	1	1	✓	12	N.D.
4	✓	✓	1	✓	12	detected
5	✓	✓	✓	✓	9	N.D.
6	✓	✓	✓	✓	6	N.D.
7	✓	✓	✓	✓	3	N.D.

N.D. = no detected.

Figure C1. Headspace GC Spectrum of Entry 4.

3.) The species in the "photocatalytic stage" cycle should all be numbered (not only the steps), similar to the so-called "stoichiometric phase" cycle above. The stoichiometries must be checked, the total charges of all complexes given, ligands must be specified in the caption of Fig. 5 or the figure itself (e.g. L = ? n = ?).

It is not always clear why species involved in the "stoichiometric phase" look so different as those in the second (catalytic) cycle below, after the reduction of hydrogen to hydride by the excited state Cu(I) complex therein. E.g. in "E", only N is shown to coordinate to copper, while in the cycle below, both N and O coordinate (if L is anionic, what is then index n? if L is neutral, is the charge correct?). The formal Cu(III) species is charged in the second cycle but uncharged in the first etc. Without exactly specifying all items, it is difficult to follow the proposal. Step iv in the first cycle has the acetate appearing as a ligand. Where does it come from as L is apparently not acetate (line 242)?

Our reply: Following Reviewer 3's suggestion, we have labeled all of the species in both "stoichiometric phase" and "photocatalytic stage". We really appreciate Reviewer 3's suggestions regarding the geometry and stoichiometry in the catalytical cycles. Unlike palladium, which has the fairly well-defined geometries, copper species are much more complicated. In our system, reactant *NH* sulfoximine, product *N*-Ar sulfoximine, solvent ethanol, along with anion from copper source, $CF_3CO_2^-$, are all possible ligands for copper in both cycles. This is why we used Ln to refer to the ligands and their number. To make the mechanism more concise, we changed all the species to [Cu] indicating oxidation state (except for the transition states) and overall charge.

Furthermore, following Reviewer 1's suggestions, we conducted DFT calculations, and modified the previous proposed mechanism. From the computational studies, the geometries of the copper species, especially the geometries of the potential photocatalysts, were predicted. With the improvement we made, which is highlighted in yellow, we hope the proposed mechanism could be easier to follow.

4.) What are the complex formation constants for A' and B' ? Is the equilibrium shown to the left in the second cycle in Fig 5 between Cu(I) (1)2 and Cu(I) (3)2 (a consistent numbering is needed

again!) possible at all ? How about having a Cu species with mixed ligands ? Is A' indeed transforming through ligand exchange into B' ? And how fast is this process?

Our reply: We really appreciate Reviewer 3 for point out the possible role of Cu(I)(1)(3) (E') with mixed ligand. From the experimental point, we have synthesized Cu(I)(1)₂ (C') and Cu(I)(3)₂ (D'), and fully characterized them in SI. However, the independent synthesis of Cu(I)(1)(3) (E') was not feasible. To answer Reviewer 3's questions regarding the role of Cu(I)(1)(3) (E'), the ligand exchange process, and related questions, we have performed the DFT calculation of thermodynamic cycles of Cu(I)(1)₂ (C'), Cu(I)(3)₂ (D'), and Cu(I)(1)(3) (E'), which are summarized in Figure S8 in SI. Interestingly, Cu(I)(1)₂ (C') is the most stable intermediate, but it was demonstrated to be an inactive PC. Cu(I)(3)₂ (D') is 13.9 kcal/mol higher than Cu(I)(1)₂ (C'), but it was demonstrated to be an active PC. The stability of the intermediate with mixed ligands, Cu(I)(1)(3) (E'), is in the middle. From an energetic standpoint, an equilibrium will populate Cu(I)(1)₂ (C') and Cu(I)(1)(3) (E') via ligand exchange at room temperature. The latter is likely the actual PC in the photocatalytic cycle. Again, the mechanism in Figure 5 has been modified on the basis of these DFT calculations.

Figure C2. Thermodynamic Cycles of Cu(I)(1)₂, Cu(I)(3)₂, and Cu(I)(1)(3)

5.) How to explain the induction stage and the subsequent dormant phase? The authors assume that ligands 1a (in contrast to 3aa) do not form photocatalytic copper complexes based on a control experiment (Table S2) and their Stern-Volmer plots of fluorescence quenching. However, the Stern-Volmer constant for complex A' is not zero and the complex definitely also absorbs visible light (Fig. 4).

Our reply: From Figure 4, the Stern-Volmer constant for A' is 1.04, very close to 1, indicating A' was not be quenched by NH sulfoximine 1a within the error of the experiment. Therefore, it

is unlikely that **A'** serves as photocatalyst under our photocatalytic conditions. In this context, the induction period indicates the very early stage of the stoichiometric cycle, when *N*-arylated product has to accumulate to serve as ligand for the photocatalytic cycle. The long dormant stage likely arises from a combination of a rate-limiting disproportionation in the stoichiometric phase and slow ligand exchange process to displace the *NH* sulfoximine from the very stable complex Cu(I)(1)_2 to generate the actual photocatalyst.

I do not understand why the absence of an external oxidant should result in the curiously long "dormant phase" under irradiation. Wouldn't it be more plausible to assume that some transformation of more active copper complexes has to occur first, before the second self-accelerating autocatalytic phase (this time by product **3aa** rather than reactant **1a**) sets in ?

Our reply: In the absence of oxidant, the top catalytic cycle is not closed, so the catalyst cannot turn over.

The photocatalysis could therefore also be by complexes formed from the *NH* rather than *NR* species alone, i.e. might involve also the reactant **1a** (!) and not only the product **3aa** as ligands. This would also explain the self-decelerating "induction stage" (Fig.3a). See for a recent paper which should be cited and discusses an autocatalysis by the reactant: ChemPhysChem 2020, 21, 1775.

Our reply: We thank the reviewer for this interesting suggestions. From calculations, the actual photocatalyst does likely incorporate both *NH* and *NR* species $[\text{Cu(I)(1)(3)}]$ as the reviewer suggests (see Figure C2 and S8 for details). That said, the adduct from just the *NH* species $[\text{Cu(I)(1)}_2]$ was tested, but did not exhibit any emission quenching by reactant or solvent. The paper (ChemPhysChem 2020, 21, 1775) was cited as Ref. 48 in the modified manuscript following reviewer 3's suggestion.

6.) I also wondered that the authors did not try to do an experiment with an UV lamp to see whether the rates and yields increase, seeing that absorption in the visible region is very small. It is really difficult to understand this, especially as alone the Cu(II) complex (Fig. 4 and 5) is apparently absorbing visible light to a significant extent. Does the reaction e.g. still proceed when the absorption of visible light is prevented under UV light irradiation (by use of appropriate filters)? Such an experiment would be really helpful.

Our reply: This is a very interesting experiment suggested by Reviewer 3. Several control reactions under irradiation of 4 W 254 nm UV lamp have been conducted, and the reaction was monitored by HPLC in two hour increments. The desired product **3ab** was generated sluggishly, and reached 15% after 48 h. Some byproducts were also observed, with the four major byproducts (**bp1-bp4**) being isolated and characterized by NMR and HRMS. We proposed that the reaction could proceed under the irradiation of UV light, but the *N*-arylated product **3ab** also decompose simultaneously. Meanwhile, the free sulfoximine **1a** does not decompose under UV irradiation. When the light source was changed to a 4 W white lamp with a filter (only light above 400 nm can pass through) from ambient light under otherwise identical conditions, these byproducts were not observed. This finding is consistent with the impetus to use lower energy visible light to avoid the attendant nonspecific bond cleavages that higher energy UV light can cause.

Scheme C1. Copper-Catalyzed Arylation of Free Sulfoximine **1a with **2b** under Irradiation of UV.**

7.) Shouldn't the actual resting state be the Cu(I) ground state, rather than a Cu(II) species resulting from the redox reaction? The reality of Cu(III) oxidation state has been very recently questioned, that paper needs to be cited and discussed and the proposed mechanism perhaps revised, see: JACS 2019, 141, 18508. This could also affect the disproportionation of Cu (II) into Cu(I) and Cu(III), which the authors postulate in Fig. 5.

Our reply: The UV spectrum of the reaction mixtures is shown in Figure 4a, and there clearly was a band between 700-800 nm, the characteristic band of Cu(II) species, indicating the resting state of the catalysis is probably Cu(II). The paper regarding reductive elimination of Cu(III) to form C-C bond (JACS 2019, 141, 18508) is cited as Ref. 51 in the revised manuscript. The reviewer brings up an excellent point about the electronic configurations of "Cu(III)" which might be more accurately described as Cu(II)-radical complexes. The proposed mechanism has been modified in this light based on new DFT calculations. Please see the answers to Question 6 of Reviewer 1 and Questions 3&4 of Reviewer 3 for details. An excellent example of this point is G' in the revised mechanism (Fig 5). Indeed, in the calculations, oxidation state is not specified, only charge and spin. We did examine multiple spin states and the structures presented are those that are lowest energy across these considerations. The Cu(III) designation is used as *formalism* to focus attention of the key substrate redox changes and to align with generally-used conceptualizations of the Chan-Lam mechanism. Otherwise, it is rather confusing since the spin is distributed across multiple atoms. Drawing accurate and understandable of such species is difficult. To help clarify this issue, "formal Cu(III)" is used and is accompanied by the suggested reference.

REVIEWERS' COMMENTS

Reviewer #1 (Remarks to the Author):

The authors have adopted most suggestions, made necessary revisions, and answered most reviewers concerns. I feel the manuscript should be accepted for publication.

Reviewer #3 (Remarks to the Author):

I have rarely seen such a meticulously and insightful revision prepared by authors! All points from my earlier report have been addressed. This paper is important and contains several thought-provoking ideas. I strongly recommend its publication in the present form.